# NOISE-ROBUST PREFERENCE LOSSES FOR DEEP REGRESSION MODELS

## ABSTRACT

Deep regression models are widely employed for tasks such as pricing and forecasting. In industry applications, it is common for analysts to adjust model outputs before they are deployed in commercial products. These adjustments, which we name "analyst influences", not only ensure the quality of the final products but also provide training data to improve model performance over time. However, due to the huge volumes of data, analyst influences can be applied broadly and can lack precision, hindering training effectiveness. To resolve the issue, we propose a novel method: *Preference Learning from Analyst Influence*, which creates a weighted loss function that explicitly accounts for the relative quality levels of the training samples in comparison to model outputs. This approach effectively mitigates the impact of coarse training instances. Our extensive experiments on real-world data drawn from airline revenue management demonstrate that the proposed method not only enhances pricing stability but also improves alignment with analyst influences compared to baselines.

## 1 INTRODUCTION

Deep regression models are widely used in many areas of study such as dynamic pricing (Ye et al., 2018; Kolbeinsson et al., 2022b; Zhang et al., 2019), asset pricing (Chen et al., 2024a), and a wide range of forecast tasks (Fernández-Delgado et al., 2019). In these applications, leveraging deep learning methods is especially important because of the high cost and low availability of analysts with the necessary expertise and specialty skills. However, due to the complexity of real-world data, deep regression models can sometimes produce inaccurate outputs, necessitating human intervention post-training. The human interventions not only safeguard the application but also provide new and additional training data for future training to improve the models.

Existing work explored learning directly from human annotation for improving performance and serviceability using reinforcement learning or supervised fine-tuning, known as human preference optimization (Ouyang et al., 2022; Bai et al., 2022a; Rafailov et al., 2024). However, these methods only apply to probabilistic models and not to regression models, necessitating specialized preference learning methods for deep regression models.

In regression tasks, the human annotation process typically requires analysts with a high level of knowledge and sophistication due to the requirement of mathematical and domain knowledge. This makes meticulous intervention challenging for large quantities of data, and analysts will have to apply the intervention to a large number of instances based on filtering queries; we name this type of annotation "analyst influence".

While analyst-influenced data is valuable for training, broadness in analyst influences makes them coarse in quality. Consequently, direct training on influenced data can diminish the sensitivity of models and potentially erase learned conditional features over generations of retraining. This necessitates a tailored method to learn from analyst influences while avoiding sensitivity loss.

To address these issues, we explore a novel loss function depending on the **relative quality** of the influenced data, defined as the probability of the influenced data being more accurate compared to the training model output. Intuitively, the higher the relative quality of the influenced data is, the better it will facilitate the training of the models, and vice versa. During training, our approach incorporates higher weighted loss when the model output is of lower accuracy and restricts the weighted loss on

lower quality data toward the end of training avoiding the problem of coarse influence. We name this approach as **P**reference **L**earning from **A**nalyst **I**nfluence (PLAI). Our method is distinct from existing loss methods for noisy labels (Wang et al., 2019; Zhang & Sabuncu, 2018; Ghosh et al., 2017; Song et al., 2022) since existing works focus on classification tasks and cannot be adopted for regression tasks.

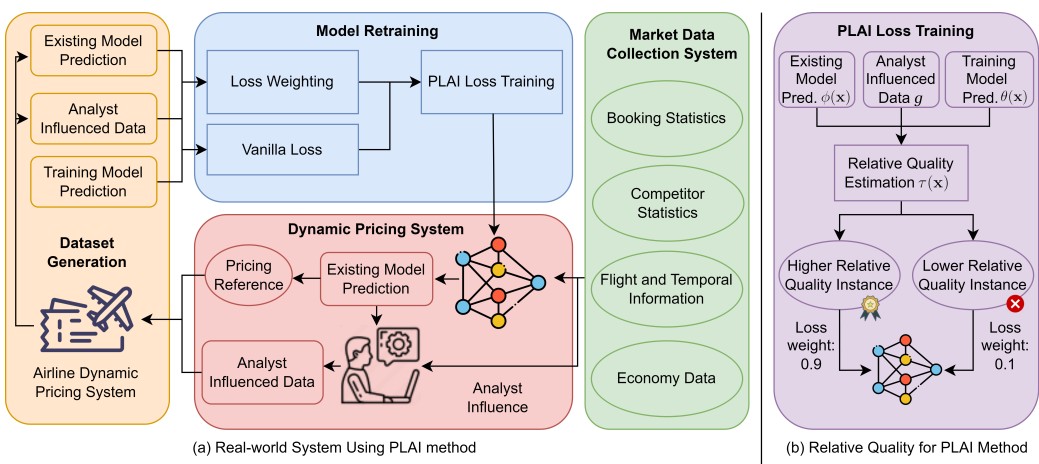

(a) Real-world System Using PLAI method        (b) Relative Quality for PLAI Method

Figure 1: Illustration of the PLAI method and how it is applied to a real-world application of dynamic pricing in airline revenue management.

To validate our hypothesis, we performed extensive experiments on dynamic pricing for airline revenue management with real-world data from Osprey Airline (Pseudonym) where the goal of revenue management is to align the model pricing strategy with analyst influence while imitating the trend of historical data for flights not being influenced. The visualized flow chart is shown in Figure 1 where the models are trained using imitation learning of historical accumulation of analyst influence. We establish evaluation metrics for both alignment with analyst preference and price stability. Our PLAI loss approach consistently outperformed the baseline methods in all evaluations.

Our contributions are the following:

- We discover the relative quality phenomenon for deep regression tasks with analyst influence and theoretically establish a method for estimating relative quality.

- Built upon our theoretical discovery, we introduce a novel method, Preference Learning from Analyst Influence (PLAI), that effectively trains deep regression models on coarse data with analyst influence.

- We show, through extensive experiments, the PLAI method outperforms our baselines for various evaluation metrics.

## 2 RELATED WORK

### 2.1 LEARNING WITH HUMANS

Learning with humans is a subfield of human-in-the-loop machine learning (Mosqueira-Rey et al., 2023; Monarch, 2021) which involves humans in helping not only contribute to safeguarding machine learning models in the production environment but also provide labeled training data. Learning with humans is especially important in real-life industry settings where data acquisition can be challenging while machine learning models typically require monitoring; examples include manufacturing (Bhattacharya et al., 2023), autonomous vehicle (Wu et al., 2023), and healthcare (Bakken, 2023).

## 2.2 PREFERENCE LEARNING

In natural language processing, preference learning is used to align large language models toward human preference. In the field of natural language processing, preference learning is pioneered by Reinforcement Learning from Human Feedback (Ouyang et al., 2022) which leverages human annotation to train a reward model and uses reinforcement learning algorithm Schulman et al. (2017) to train a target model to align with human preferences. Later works use artificial intelligence models for the purpose of annotation, namely reinforcement learning from AI feedback, to reduce human effort involved (Bai et al., 2022b; Lee et al., 2023). Besides reinforcement learning approaches, Rafailov et al. (2024); Azar et al. (2024) provides simplified methods for preference learning without the need of training a reward model.

# 3 BACKGROUND AND CONTEXT FOR APPLICATION AREA

## 3.1 DYNAMIC PRICING FOR AIRLINE REVENUE MANAGEMENT

Dynamic pricing is the current state-of-the-practice in airline revenue management, involving the continuous adjustment of ticket prices based on real-time demand, booking patterns, and market conditions to maximize revenue (Van Ryzin & Talluri, 2005; Belobaba et al., 2015; Kolbeinsson et al., 2022a). Unlike traditional static pricing, where ticket prices are fixed or adjusted infrequently, dynamic pricing enables airlines to respond quickly to changes in demand, such as increasing prices as seats fill up or offering discounts when demand is low. This adaptability helps airlines manage seat inventory more efficiently, capture consumer surplus, and increase overall profitability by selling the right seat to the right customer at the right price at the right time. Conventionally, the airline industry has adopted traditional approaches for revenue management and is only recently adopting deep learning methodologies to improve the accuracy and granularity in their pricing models.

## 3.2 BID PRICE PREDICTION

Bid price prediction is a critical task in airline revenue management aimed at helping airlines increase revenue from ticket sales. Conceptually, the bid price represents the marginal opportunity cost of a seat on a flight at an observed date. Bid prices are considered the lower bound of seat prices and help airlines protect seats for higher-paying business customers who are less sensitive to pricing and often book closer to the departure date. An accurate estimate of bid prices dynamically generated during the booking process allows an airline to estimate if they should sell a seat at a given price or hold to increase the price.

At our partner airline, bid prices are calculated daily for each flight, with updates available up to 384 days before the departure date to ensure accurate pricing. The predictions are based on available features from Osprey Airlines (pseudonym) and the data pipeline from our company. These features include time-related factors (such as observation date, departure date, and day of the week), geographical information (origin and destination), load factor [1], booking history, and statistical data on both internal and external competitor flights.

## 3.3 ANALYST INFLUENCE

At our collaborating company, analysts from airlines use analytical tools to monitor the production model and apply influences to modify flight prices. These analyst influences exist historically for the tasks of bid price prediction. There are three types of influences:

**Market Constraints**: Restrictions on the bid price based on the load factor.
**Price Relative**: An increase or decrease in the bid price by a certain percentage.
**Price Absolute**: An increase or decrease in the bid price by a specific amount.

These influences can be applied based on filters such as load factor, flight number, departure dates, observation dates, or regions to target large groups of flight prices. A significant percentage of flight prices are influenced; during a 60-day period starting from January 16, 2024, a total of 12,399,775

---

[1]Load factor of a flight is defined as the percentage of the seats being sold on the flight.

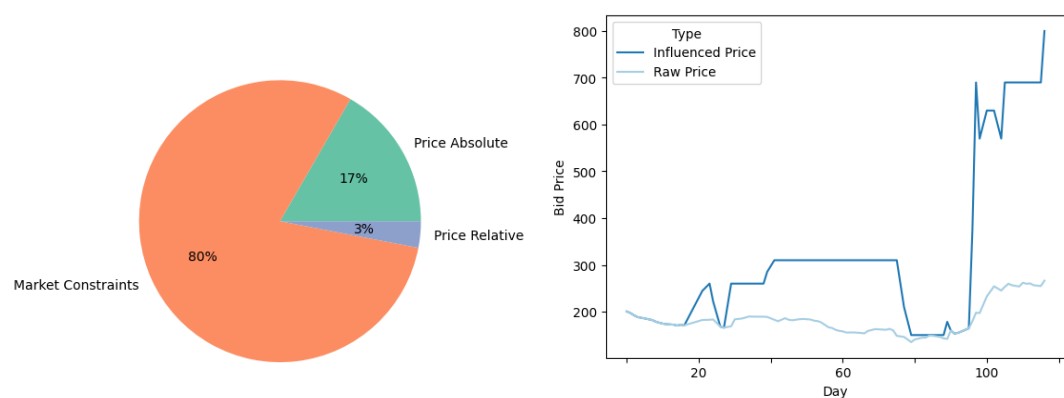

(a) Distribution of different types of analyst influences. Market constraints are the most popular methods since they are independent of the model's deployment.

(b) Analyst influence increased the prices for multiple periods since the popularity of the flight was not recognized by the production model.

Figure 2: While analysts help correct model output and adjust flight prices based on macro market conditions, analyst influences are broad and coarse in quality due to limited capacities from analysts.

bid prices were in production, with 6,810,134 prices (54.92%) affected by influence, and 5,244,568 prices (42.30%) experiencing an influence larger than 15 percent of the actual prices.

However, analyst influences also come with challenges for model training. Due to the limited availability of analysts, these influences are often applied to a large number of flight prices. From 2019 to 2024, the analyst influences from Osprey Airlines affected an average of 8,401 flight prices. As a result, the adjusted prices are not fine-grained and may not retain correlations with crucial pricing determinants. A sampled influenced flight price is shown in Figure 2.

## 4 PREFERENCE LEARNING FROM ANALYST INFLUENCE

### 4.1 COMPARISON TO EXISTING METHODS

In this paper, we address the challenge in regression tasks involving analyst influence; where the influence while accurate in aggregate, can be coarse at the individual record level. Despite being a common problem for tasks such as dynamic pricing and forecasting, very little academic research directly tackles this issue. The closest problems are human alignment optimization in natural language processing (Ouyang et al., 2022; Bai et al., 2022a; Rafailov et al., 2024) and methods addressing noisy labels (Wang et al., 2019; Zhang & Sabuncu, 2018; Ghosh et al., 2017; Song et al., 2022). However, human alignment optimization typically relies on assumptions such as the Bradley-Terry model (Bradley & Terry, 1952) for human preference; on the other hand, the noisy labels methods only address classification tasks. In both cases, the solutions target probabilistic models and cannot be adopted for regression models.

### 4.2 NOISE-ROBUST LOSS WEIGHT VIA RELATIVE QUALITY

A simple approach for training models for the task of bid pricing with analyst influence is *supervised fine-tuning*, i.e., reducing the discrepancy between analyst-influenced price and the price provided by the previous ML model. However, due to the coarse nature of analyst influences (as shown in Figure 2), influenced data can erase some sensitive dependencies the model learned, resulting in poor performance. To address this issue, we introduce *relative quality*, a metric that computes the probability that the training data are more accurate compared to the output of the training model. Relative quality can be utilized as a weighting mechanism during training, helping mitigate the negative impact of coarse influences.

Given a group of training samples with varying input conditions but identical prices due to coarse analyst influence, the use of relative quality as a weighting factor can reduce the loss as the model's mean output approaches the influenced price, helping retain the model's sensitivity to the original input conditions. However, while in the early stage of the training, the model producing lower quality output will receive a full loss update due to high relative quality, facilitating faster training. A schematic of our approach with loss weights modified using relative quality is in Figure 1b.

We formally define the problem as follows, using the same notation as in Figure 1b. We denote the input data by $x \sim \mathcal{X}$, the model being trained (with analyst influence) by $\theta$, and the previous production model (prior to any analyst influence) by $\phi$. We assume the analyst influence on price follows a Gaussian distribution, and the actual recorded influenced price (ground truth) $g$ as

$$g \sim \mathcal{G} = \mathcal{N}(\mu(x), \sigma(x)). \tag{1}$$

Here $\mu$ is an omniscient model[2] and $\sigma$ is the standard deviation of the recorded influenced prices given $x$. We further assume the probability of analyst influence being better than the previous model output is $\delta$.

The goal of this theoretical analysis is to utilize *relative quality* $\tau(x, \theta, g)$ for loss weights during the training

$$\mathcal{L}_{weighted} = H(\tau(x, \theta, g))\mathcal{L}_{regression} \tag{2}$$

where $H$ is a penalty function that can be tuned because it is unclear whether directly multiplying the loss by the relative quality is optimal.

### 4.3 ESTIMATING RELATIVE QUALITY

Although it is typically impossible to calculate relative quality directly in most situations, we demonstrate that it can be estimated for regression tasks that involve analyst influences. We formally define relative quality as:

$$\tau(x, \theta, g) = P[\mathcal{G}(\theta(x)) < \mathcal{G}(g)] \tag{3}$$

Because we assume $\mathcal{G}$ is Gaussian, this expression can be simplified to

$$\tau(x, \theta, g) = P[|\mu(x) - \theta(x)| > |\mu(x) - g|] \tag{4}$$

Further, since $g$ is sampled from $\mathcal{G} = \mathcal{N}(\mu(x), \sigma(x))$, we have $g = \mu(x) + \epsilon\sigma(x)$ where $\epsilon \sim \mathcal{N}(0, 1)$,

$$\tau(x, \theta, g) = P\left[\frac{|\mu(x) - \theta(x)|}{\sigma(x)} > |\epsilon|\right] = 2F\left(-\frac{|\mu(x) - \theta(x)|}{\sigma(x)}\right) \tag{5}$$

Where $F(x) = \frac{1}{\sqrt{2\pi}} \int_{-\infty}^{x} e^{-\frac{u^2}{2}} du$ is the cumulative distribution function of the normal distribution. In the above expression, $\mu(x)$ can be estimated by $g$ because analyst influences are broad and coarse but not out of range.

We now expand $\delta$, the probability that the analyst influence is better than the output of the prior model ($\phi$) for estimation of $\sigma(x)$ as:

$$\delta = E_{x' \sim \mathcal{X}}[\mathcal{G}(\phi(x')) > \mathcal{G}(g)] = E_{x' \sim \mathcal{X}}\left[2F\left(-\frac{|\mu(x') - \phi(x')|}{\sigma(x')}\right)\right] \approx 2F\left(-\frac{|\mu(x) - \phi(x)|}{\sigma(x)}\right). \tag{6}$$

This gives

$$\sigma(x) \approx -\frac{|\mu(x) - \phi(x)|}{F^{-1}(\frac{\delta}{2})} \approx -\frac{|g - \phi(x)|}{F^{-1}(\frac{\delta}{2})} \tag{7}$$

We then achieve a final form of relative quality after applying equation 7 to equation 5:

$$\tau(x, \theta, g) \approx 2F\left(F^{-1}(\frac{\delta}{2}) \cdot \frac{|g - \theta(x)|}{|g - \phi(x)|}\right) = 2F\left(c \cdot \frac{|g - \theta(x)|}{|g - \phi(x)|}\right) \tag{8}$$

---

[2]We consider this to be a fine-grained analyst's influence, *i.e* the influence an analyst will make given infinite time.

## 4.4 PLAI LOSSES

The estimate of relative quality contains the function $F(c \cdot z)$ (equation 8), which does not have a simple form. However, we combine it with the penalty function $H$ to tune a joint penalty function $\bar{H}(z) = H(F(c \cdot z))$ such that the weighted loss takes the form

$$\mathcal{L}_{weighted} = \bar{H}(r)\mathcal{L}_{regression} \tag{9}$$

where $r = r(x, \theta, \phi, g) = \frac{|g-\theta(x)|}{|g-\phi(x)|}$ is the inner relative quality.

The inner relative quality $\frac{|g-\theta(x)|}{|g-\phi(x)|}$ has an intuitive interpretation. The numerator $|g - \theta(x)|$ can be seen as the inverse of the quality of the model, with a higher numerator indicating a lower quality of the model. On the other hand, the denominator $|g - \phi(x)|$ is a scaling factor to ensure the value of inner relative quality if invariant with the magnitude of output value.

We provide three implementations of the $\bar{H}$ function as shown in Equation 9. As a remedy to the estimation errors, we design the weight $\bar{H}$ of PLAI losses to have a maximum value of 1 as the inner relative quality goes to infinity.

In particular, inspired by existing works exploring the sigmoid function for noisy labels (Ghosh et al., 2015; Chen et al., 2024b), we define **Sigmoid PLAI loss** as

$$\mathcal{L}_{sigmoid} = -sigmoid(\alpha(r-1))|\theta(x) - g| = -\frac{1}{1 + \exp(-\alpha\frac{(|g-\theta(x)|-|g-\phi(x)|)}{|g-\phi(x)|})}|\theta(x) - g| \tag{10}$$

Alternatively, we introduce **Focal PLAI loss** based on focal loss (Lin, 2017) which imposes higher penalties for training instances with moderate relative quality

$$\mathcal{L}_{focal} = -(\frac{r}{r+1})^\gamma|\theta(x) - g| = -(\frac{|g-\theta(x)|}{|g-\theta(x)| + |g-\phi(x)|})^\gamma|\theta(x) - g| \tag{11}$$

As shown in Figure 3, the focal PLAI loss has a harsher penalty when the inner relative quality is near 1 as $\gamma$ increases; on the other hand, sigmoid preference overall has a lower loss penalty than focal PLAI loss and has a stable relative quality value at $0.5$ when the inner relative quality is near 1.

Lastly, we introduce a simple approach that uses the clip function to limit the inner relative quality into the range of $[0, 1]$

$$\mathcal{L}_{clip} = -clip(r, 0, 1)|\theta(x) - g| = -clip(\frac{|g-\theta(x)|}{|g-\phi(x)|}, 0, 1)|\theta(x) - g| \tag{12}$$

## 5 EXPERIMENT

### 5.1 SETUP

**Dataset.** Our dataset is constructed using cleaned and processed input features and bid price data from Osprey Airlines (pseudonym) together with data from third-party vendors. A non-exhaustive list of input features includes the observation date, departure date, day of the week for departure, departure time, origin, destination, load factor, previous booking information, and statistical information of internal and external competitor flights.

We split the dataset by date to create training, validation, and test sets. The departure dates range from November 29, 2019, to January 15, 2025. The training dataset includes departures from November 29, 2019, to February 15, 2024, the validation set covers February 16, 2024, to April 6, 2024, and the test set spans April 7, 2024, to January 15, 2025. To avoid data contamination, we ensure that observation dates in the test set do not overlap with the departure dates used in the training and validation sets. Each individual flight has 384 days of bid price history, resulting in a dataset with over 100 million bid prices.

**Model.** We use a model consisting of one convolutional layer and 12 gated linear unit (GLU) layers, similar to those used in modern transformers. The convolutional layer is specifically designed to

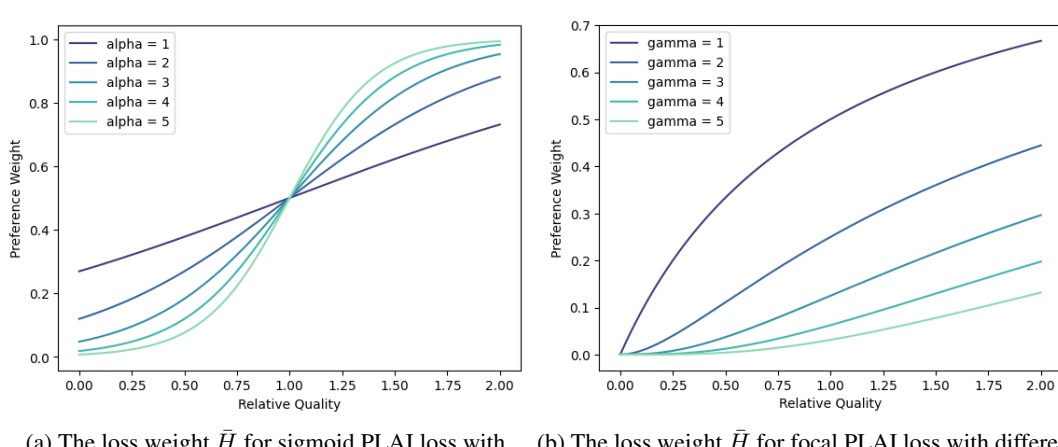

(a) The loss weight $\bar{H}$ for sigmoid PLAI loss with different $\alpha$.

(b) The loss weight $\bar{H}$ for focal PLAI loss with different $\gamma$.

Figure 3: We show the preference weight $\bar{H}$ of different PLAI loss. The focal PLAI loss has a harsher penalty as $\gamma$ increases. On the other hand, sigmoid PLAI loss has an overall stable penalty. The hyperparameter $\alpha$ affects mainly the extremities.

learn historical booking features, while the GLU layers process the output of the convolutional layer along with other input features. The GLU layers have a hidden dimension of 256, an intermediate dimension of 768, a dropout rate of 0.1, GELU activation, and RMS normalization. The total number of parameters in the model is 3.7 million. We use the AdamW optimizer with a learning rate of 5e-4. During training, we employ a batch size of 128 flights, which is equivalent to 49,152 bid prices.

**Baselines.** We consider different regression loss functions as the baselines of our experiment. These include MAE, MSE, Huber, and logcosh losses.

## 5.2 EVALUATION METRICS

The evaluation is a challenging problem for dynamic pricing, as none of the evaluation metrics can fully encapsulate the performance of a model. To provide comprehensive metrics for the needs of our partner airline, we consider the two factors of evaluation:

- **Stability**: A pricing model in a commercial system needs to show stability in output to avoid mayhem. We use two methods to evaluate the stability of model outputs. First, we assess the proximity of the model output to the recent production pricing. This helps determine if the model can successfully imitate the influenced output as well as desirable traits of the previous production model. We use Mean Absolute Error (MAE), Mean Squared Error (MSE), Mean Absolute Percentage Error (MAPE), and Root Mean Squared Error (RMSE) to evaluate the effectiveness of imitation learning. On the other hand, due to the seasonality of the airline industry, seasonality is a crucial perspective for pricing models to learn. We provide seasonality analysis by aggregating the pricing by departure week to observe whether the trained model learned the seasonal trend.

- **Analyst Alignment Metrics**: We evaluate whether the model outputs are moving in the direction of influenced prices compared to the outputs of the production model (which were also influenced by analysts). We use accuracy in confusion matrices to assess the success of the model in learning analyst influences.

## 5.3 RESULTS

**Analyst Preference Alignment:** To evaluate the effectiveness of the model in learning the analyst preference, we evaluate whether the output difference of the trained model compared to the previous production model is in the same direction as the analyst's influence. We classify influence into three categories: price increase of 15% or more, price decrease of 15% or more, and price unchanged with a price change of 15% or less. Then we compare the training model against the previous production

| Method | Accuracy (Percentage) |
|---|---|
| MAE Loss | 50.34 |
| MSE Loss | 48.96 |
| Huber Loss | 49.06 |
| Logcosh Loss | 50.10 |
| MAE: 25% weight on influenced | 47.27 |
| MAE: 50% weight on influenced | 47.27 |
| MAE: 75% weight on influenced | 46.01 |
| Simplified Sigmoid PLAI Loss | 49.89 |
| Simplified Focal PLAI Loss | **52.96** |
| Simplified Clip PLAI Loss | 50.52 |
| Sigmoid PLAI Loss | 51.29 |
| Focal PLAI Loss | 50.43 |
| Clip PLAI Loss | 51.98 |

Table 1: Influence Accuracy for Different Loss Functions

model and categorize the model change in the same way. We compute the accuracy by comparing the model change against the influence change. We use the accuracy to evaluate the effectiveness of the model in learning the analyst preference. As shown in Table 1, the experiment results suggest that even the Focal PLAI loss, the lowest-performing PLAI loss, is stronger than the highest-performing baseline emphasizing the effectiveness of the PLAI framework. On the other hand, the performances among PLAI losses also vary, combined with the shape analysis of the loss weight and the relative quality, we observe that a closer to linear relation between loss weight and relative quality improves the accuracy of preference learning.

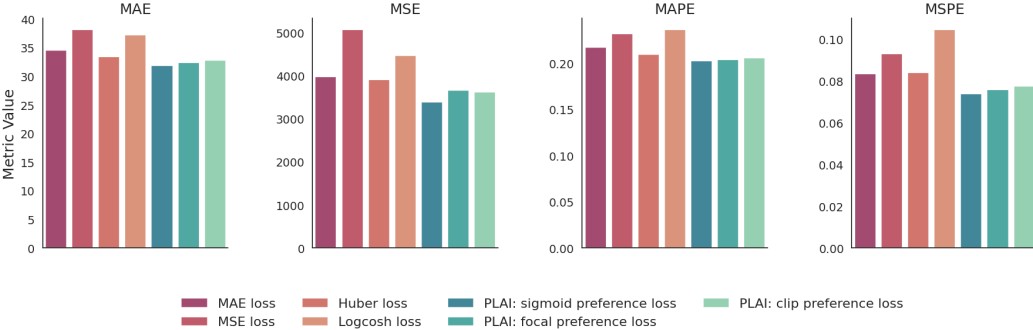

Figure 4: Comparison of prices from multiple imitation learning methods, including the proposed PLAI approach, against the ground truth. Observe that the trends are similar across different evaluation metrics, with PLAI outperforming existing approaches. *Best viewed in color.*

**Imitation learning.** The results of imitation learning are shown in Figure 4. With a few exceptions, the rank of different imitation metrics stays the same across different metrics; therefore, for the model experimented, there is no distinct performance difference between higher-priced flights and lower-priced flights as they would result in discrepancies between MAE and MSE or between MAE and MAPE. Across all different evaluation metrics, all PLAI losses outperform the baselines. The result demonstrates the stability of PLAI losses with their ability to imitate the previous ground truth. Among PLAI losses, the sigmoid PLAI loss shows the highest performance in the experiment.

Combined with the influence alignment result, we observe that PLAI methods exceed the performance of the baseline in both evaluations. Therefore, regardless of the $\bar{H}$ formulations, PLAI methods can align better to analyst preference while improving on imitating the ground truth $g$.

**Domestic vs international.** During the regional evaluation, the PLAI methods show significant performance improvement on domestic flights while having competitive performance on international flights as shown in Figure 5. At our partner airline, domestic flights account for 86 % of all flight prices and are sensitive to diverse dynamic features such as booking and competition; therefore,

with higher performance in domestic flights, the PLAI losses are successful in preserving sensitivity to dynamic features, leading to higher-quality domestic flight price output.

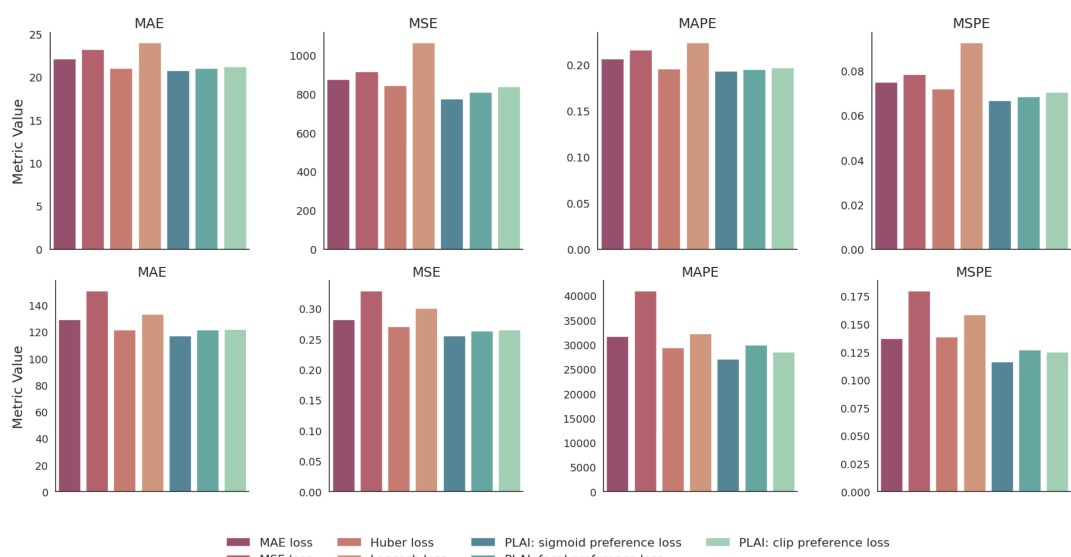

Figure 5: Imitation evaluation with the domestic and international flights separated.

**Seasonality** As shown in Figure 6, we report the analysis of seasonality for the trained models. Overall, we see prices increase in July for the Summer travel season and November, December, and January for the holiday seasons. With the exception of the *logcosh* loss which has a price hike in August, all trained models have output average prices close to the ground truth after aggregation by week.

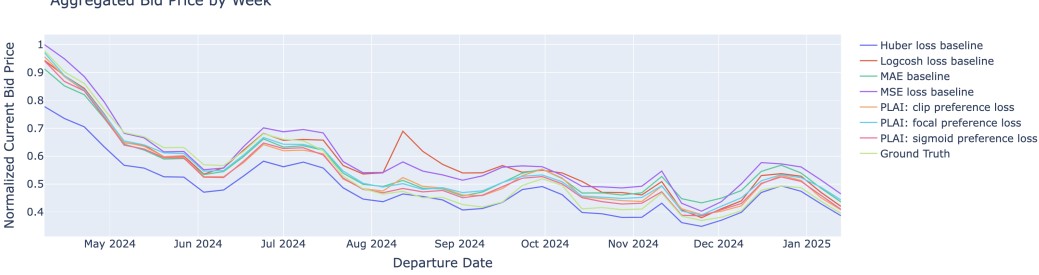

Figure 6: We show the average model output prices by departure date of different methods. This shows the ability of models to learn the seasonality in flight pricing.

### 5.4 ABLATION STUDY

**Alternative Penalties:** As PLAI losses can be considered as penalties on low-relative-quality records with analyst influences, we perform an ablation study to compare PLAI losses to alternative penalties on influenced prices. We consider two alternative approaches: First, we use MAE loss with 25%, 50%, and 75% learning rates on analyst-influenced records only. This represents a simpler penalty on influence prices without the PLAI method to distinguish between records with high and low relative quality. Second, we replace the scaling factor $|g - \phi(x)|$ with $g * \zeta$ as an alternative scaling method, $.i.e\ r = \frac{|g-\theta(x)|}{g*\zeta}$. This reduces one parameter of the inner relative quality which we refer to as *simplified PLAI loss*. In our experiment, we use $\zeta = 0.1$. In both stability evaluations as shown in Figure 7 and alignment evaluations as shown in Table 1, PLAI losses outperform MAE losses with reduced loss weight on records with analyst influence, demonstrating the effectiveness of formulation in inner relative quality beyond penalties to analyst-influenced records. On the other

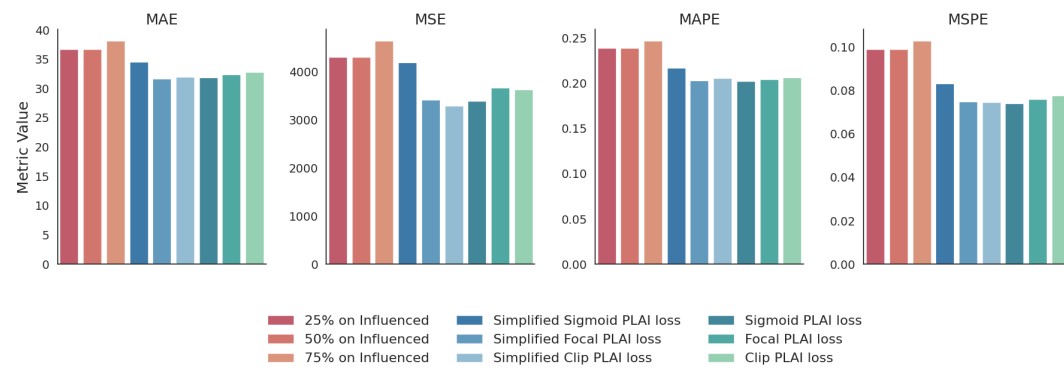

Figure 7: We compare the imitation learning results comparing PLAI losses and simplified version of PLAI losses.

hand, the performance of simplified PLAI loss is similar to the PLAI loss across different evaluations. This aligns with our interpretation of the inner relative quality showing that the alternative approach can be effective if they serve the same purpose as the original formulation.

**Hyper-Parameters:** We perform hyper-parameter studies for Sigmoid and Focal PLAI losses, as shown in Figure 8. As the hyper-parameters in both PLAI losses represent penalties for records with lower relative quality as shown in Figure 3, we conclude that for both losses, a properly chosen hyper-parameter (penalty) can lead to higher performances.

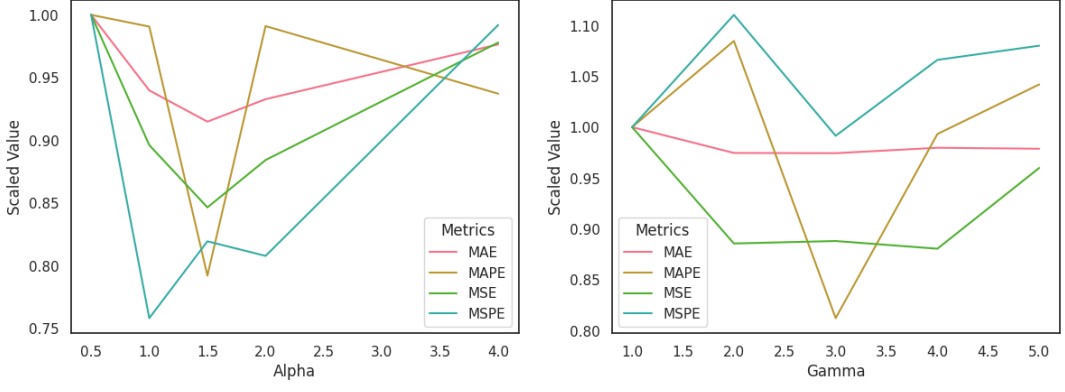

(a) The performance for Sigmoid PLAI loss with different $\alpha$.

(b) The performance for Focal PLAI loss with different $\gamma$.

Figure 8: We show the effectiveness of different hyper-parameters on the performance of sigmoid PLAI loss and focal PLAI loss scaled by the first entry. While the results of MAPE and MSPE are inconclusive due to flights with lower prices, for MAE and MSE, the performance peaks at a centered value and decreases as the hyper-parameter value is away from it.

# 6 CONCLUSION

We introduced preference learning from analyst influence, a loss method that leverages relative quality to reduce the lower-quality influenced data to hinder the model performance. We theoretically establish an estimation of the relative quality and perform experiments with multiple variations of the PLAI method. As a result, our proposed losses exceeded all baselines in both stability and alignment with the analyst influences.

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
