# OpenReview forum: "Noise-Robust Preference Losses for Deep Regression Models"
_ICLR.cc/2025/Conference — Submitted to ICLR 2025_

### Official Review · Reviewer_idp9 · 2024-10-21

**Soundness:** 2
**Presentation:** 3
**Contribution:** 1
**Rating:** 3
**Confidence:** 4

**Summary:**

This paper proposes a weighted loss function for deep regression models. The weight of the loss is determined by the quality of the so-called ‘analyst influence’ on the training data. If the coarsely adjusted analyst data performs well on the training sample, the corresponding loss weight is higher than for those samples where the fit is poorer. The quality of the analyst’s influence is assessed using the existing model.

**Strengths:**

The paper is well-organized, and the proposed loss function includes some theoretical analysis

**Weaknesses:**

W1: The proposed method lacks novelty. Weighted loss functions have been extensively studied, and the potential application and contribution of the proposed PLAI loss function are limited.

W2: The authors spend half a page explaining "airline revenue management" and "Bid price prediction," which is a digress from the main subject and does not interest most readers.

W3: There is no comparison with state-of-the-art models; only different loss functions were compared. Additionally, the proposed PLAI loss does not show significant improvement in influence accuracy. Compared to MAE loss, the proposed PLAI improves influence accuracy by only 2%.

**Questions:**

The paper reads more like an industrial report than an academic paper. The paper is not ready for publication at ICLR.

---

> ### Comment · Reviewer_idp9 · 2024-11-26
> **No rebuttal**
>
> The author did not participate in the rebuttal process. The score remains unchanged.

---

### Official Review · Reviewer_Dwzb · 2024-10-30

**Soundness:** 2
**Presentation:** 2
**Contribution:** 2
**Rating:** 3
**Confidence:** 3

**Summary:**

This paper investigates a real-world problem: dynamic pricing for airline revenue management. The authors propose the PLAI method, which is simple and easy to follow. Based on the analyst influenced data, this paper discovers the relative quality for deep regression tasks in dynamic pricing. Training with data with analyst influence, this paper conduct extensive experiments to verify the effectiveness of PLAI method,.

**Strengths:**

1. The paper investigates a real-world problem and models the problem well. The problem setting is interesting.

2. The proposed method utilizing the analyst influence data is well-motivated in this setting. The PLAI fits the practical problem.

3. Experiments on industry data are well-designed.

**Weaknesses:**

1. The problem is orginated from practical problem, as a result, the background should be stated clearly. The background is not so easy to understand in section 3.3. It is not so clear to interpret Figure 2.

2. The contribution of this proposed method is not so obvious. The proposed method seems to be a continual learning for deep regression tasks. The distinction and novelty of this paper should be emphasized.

3. Baselines only cover MAE, MSE, et al., and some basic losses; more recent works in this field should be compared to make the work more sound.

4. Experiments on open datasets should be conducted to ensure the reproducibility of this work. Table 1 should show statistical significance.

**Questions:**

1. What is the meaning of Figure 2? Figure 2(a) shows market constraints are the most popular methods, but how does this affect the proposed method? Figure 2(b) shows the influenced price is consistently larger than the raw price, so how does this affect the results?

2. In Equation(1), the author assumes that analyst influence on price follows a Gaussian distribution. Please give some justification for this assumption.

3. The author proposed three variant losses based on Equation 8. Please provide some analysis of these losses, especially how they affect the optimization process.

4. Referred to weakness, more details of experiments should be given.

---

### Official Review · Reviewer_7tF2 · 2024-11-03

**Soundness:** 3
**Presentation:** 3
**Contribution:** 2
**Rating:** 3
**Confidence:** 4

**Summary:**

The authors propose a novel method called Preference Learning from Analyst Influence (PLAI), which introduces a weighted loss function that accounts for the relative quality levels of training samples compared to model outputs. The paper includes a detailed theoretical analysis, the formal definition of relative quality, and the proposal of three PLAI loss implementations: Sigmoid PLAI loss, Focal PLAI loss, and Clip PLAI loss. The experiments validate the effectiveness of PLAI in improving model performance and alignment with analyst preferences.

**Strengths:**

1.	The paper is well-structured and clearly written. The authors have successfully communicated complex concepts in a manner that is accessible to readers.
2.	The paper raises an interesting question. How does real-world analyst intervention (or ‘analyst influence’) affect the effectiveness of model training, and how does adjusting to the level of quality of the training samples relative to the model output mitigate the impact of rough training examples.
3.	The structure and iconography of the paper is clear.

**Weaknesses:**

1.	The paper is not sufficiently experimental. For example, the richness of the data is insufficient. The paper compares PLAI with several baseline methods, but it could benefit from a comparison with other state-of-the-art methods or recent advances in robust regression techniques. This would provide a more comprehensive. understanding of PLAI's performance relative to the current research landscape.
2.	The background research for the article was insufficient and it is suggested that the literature review section should be expanded to cover more existing work relevant to the research topic. This includes recent research developments, classic papers, and high-quality work that is widely recognized in the field.
3.	The link to the experiment is not given, are the results is not verifiable.

**Questions:**

1.	It looks like the design of the function of sorts isn't very new, is it possible to make a better argument for the difference between that scenario and other approaches in other scenarios? What kind of innovative design was done for this scenario? Is there any similar approach in other scenario?
2.	The title of the article is NOISE-ROBUST PREFERENCE LOSSES FOR DEEP REGRESSION MODELS, why is only the data for the airlines used? Are there any specific scenarios where PLAI outperforms or falls short of these methods?
3.	Can the code links be given for validation and reproduction of results.

---

### Meta-Review · Area_Chair_oatJ · 2024-12-20

**Metareview:**

The authors propose a novel method called Preference Learning from Analyst Influence (PLAI), which introduces a weighted loss function that accounts for the relative quality levels of training samples compared to model outputs. While this paper has some strengths like proposing an interesting question, utilizing real-world settings, and so on, the reviewers indicate several significant weaknesses including: (1) the experiments are insufficient. (2) the introduced research background is insufficient. (3) the proposed method is not novel.

In summary, I believe this paper is still not ready for publication.

**Additional Comments On Reviewer Discussion:**

Nan (the authors do not provide feedback, and the reviewers do not discuss this paper)

---

### Decision · Program_Chairs · 2025-01-22

Reject